# Preparation and Characterization of Flavored Sauces from Chinese Mitten Crab Processing By-Products

**DOI:** 10.3390/foods12010051

**Published:** 2022-12-22

**Authors:** Ying Sun, Yan Zhou, Yanmei Ren, Jianan Sun

**Affiliations:** 1Qingdao Key Laboratory of Food Biotechnology, College of Food Science and Engineering, Ocean University of China, Qingdao 266500, China; 2Key Laboratory of Biological Processing of Aquatic Products, China National Light Industry, Qingdao 266500, China

**Keywords:** crab sauce, biochemical characteristics, enzyme hydrolysate, flavor evaluation

## Abstract

To achieve high-value utilization of Chinese mitten crab processing by-products, different types of sauces were prepared using crab legs (CLs) and crab bodies (CBs). Two processing methods, enzymatic hydrolysis and enzymatic hydrolysis coupled with the Maillard reaction, were employed to prepare these sauces. An electronic nose (E-nose) and electronic tongue (E-tongue) were used to measure the changes in the taste and odor, an automatic amino acid analyzer was used to measure the amino acid content, and a headspace solid-phase microextraction GC/MS (HS-SPME-GC/MS) was used to analyze the volatile compounds, qualitatively and quantitatively. The results showed that the sour, bitter, and other disagreeable odors of the enzymatic hydrolysis solution (EHS) were reduced following the Maillard reaction; meanwhile, the umami and saltiness were considerably enhanced. The quantity of free acidic amino acids with an umami and sweet taste in the CL and CB sauces after enzymatic hydrolysis and the Maillard reaction was substantially higher than that in the homogenate (HO). The Maillard reaction solution (MRS) produced more volatile compounds than the HO and EHS, such as aldehydes, pyrazines, ketones, etc. These compounds not only impart a unique flavor but also have antioxidant capabilities, making them a prototype for the high-value utilization of crab processing by-products.

## 1. Introduction

The Chinese mitten crab (*Eriocheir Sinensis*), which is primarily found in North Asia and the Thames Valley region, is a significant freshwater economic crab in China [1]. Due to its delectable meat, high protein content, complete and optimal amino acid composition, and a wide variety of fatty acids, minerals, and vitamins that are crucial for human metabolism, the Chinese mitten crab has a high nutritional value in the aquaculture business [2]. According to the statistical report of the Chinese Ministry of Agriculture, the production of Chinese mitten crab was 808,000 tons in 2021 [3]. The extraction of crab roe is one of the important processing methods for crabs, and the processing of crabs (crab roe processing) produces a large number of by-products, including crab legs and bodies, which account for about 65% of the weight of the crab. As a result, the comprehensive utilization of the large number of by-products generated during processing is becoming increasingly urgent.

Studies have shown that the by-products of Chinese mitten crab contain nutrients, such as protein (15~30%), minerals (30~50%), chitin (15~30%), fat (1~10%), and rich vitamins [4]. Currently, research on Chinese mitten crab by-products is primarily concentrated on chitin extraction; the utilization of other components has not received attention. Making maximum use of the protein and lipid-based components in crab by-products to generate sauces is, thus, a crucial step toward developing high-value biotransformation and the recycling of crab by-products.

In the process of the green biotransformation of aquatic by-products, the demand for seafood seasonings produced by extraction, fermentation, and enzymatic digestion has continued to grow in recent years. Compared with extraction and fermentation methods, enzymatic digestion is an important tool for modernizing the production of seafood seasonings because of its simplicity, efficiency, and ease of control [5]. However, the enzymatic digestion of crab by-products is often harsh, lacking in freshness and mellowness, and deficient in overall flavor. The Maillard reaction could be employed to solve these problems. Under the appropriate reaction conditions, the Maillard reaction can remove the bitterness while also removing other unpleasant flavors [6]. The Maillard reaction has been widely used in the preparation of meat seafood seasonings, such as the Maillard reaction system, composed of blue bass, scallop, and other seafood enzymatic hydrolysis products, and the reduction of sugar to prepare rich seafood seasonings [7,8]. However, as peptides and proteins are practically incapable of Maillard reactions, and most Maillard reactions occur when they are rich in free amino acids, good enzymatic digestion of crab by-products is required for the taste enhancement process. At present, the by-products of Chinese mitten crab have been used in the extraction of chitin and its derivatives and the production of feed, and there are few reports on the preparation of seasonings with the by-products of crab [4]. Moreover, there are few reports on the systematic study and evaluation of the effects of the enzymatic hydrolysis decoupled Maillard reaction on aquatic sauces.

In this study, two kinds of sauces were created from two types of by-products, utilizing enzymatic digestion and flavor enhancement techniques based on the Maillard reaction. These sauces were then objectively assessed using current scientific analytical methodologies. The goal was to find a novel, effective approach to using crab processing by-products, increase their overall worth, and offer theoretical backing for the high-value utilization of crab sauces.

## 2. Materials and Methods

### 2.1. Raw Materials and Enzymes

Two processed by-products of the Chinese mitten crab, crab legs (CLs), and crab bodies (CBs, crab roe removed) were purchased from the Taizhou of Jiangsu Province in China, and they were stored at −20 °C. The flavor protease, with an enzyme activity of 500 LAPU/g, and papain, with an enzyme activity of 1 × 10^5^ U/g, were used. These two kinds of enzymes were both purchased from Novozymes Biotechnology Co. Ltd. (Beijing, China).

### 2.2. Raw Material Basic Components

An atmospheric pressure electronic moisture analyzer was applied to assess the moisture content of the CLs and CBs. The AOAC-established procedures were used to analyze the basic components of both the CLs and CBs [9]. Following mineralization at 525 °C for 5 h, the total amount of ash was calculated. When calculating the protein content, the protein nitrogen content was multiplied by 6.25, and the total nitrogen was calculated using an automatic analyzer (Kjeltec TM 8400, Foss, Kongeriget Danmark). Using an acid hydrolysis method, the crude lipid analysis was carried out.

### 2.3. Optimization of the Enzymatic and Flavor Enhancement Technology

#### 2.3.1. Optimization of the Enzymatic Process Conditions

Two types of Chinese mitten crab processing by-products were used as source meterials (CLs and CBs) for enzymatic hydrolysis. The following five factors were considered during the enzymatic hydrolysis process: enzyme species (flavourzyme and papain), enzyme treatment time (0 min, 20 min, 30 min, 40 min, 50 min, 60 min, and 70 min), material-to-]. TVB-N was utilized to calculate the ideal enzymatic hydrolysis time to evaluate the liquid ratio (1:1 g/mL, 1:2 g/mL, 1:3 g/mL, 1:4 g/mL, 1:6 g/mL, 1:8 g/mL, and 1:10 g/mL), amount of added enzyme (0.5%, 1%, 2%, 3%, 4%, 5%, and 6%), and time of enzyme hydrolysis (0 h, 3 h, 6 h, 9 h, 10 h, and 11 h). The amino nitrogen concentration, as evaluated by formaldehyde titration, served as the indicator for all circumstances [10]. TVB-N was utilized to calculate the ideal enzymatic hydrolysis time to evaluate the freshness of the enzymatic hydrolysate, according to the Chinese National Standard GB/T5009.228-2016. All the experiments were repeated three times.

#### 2.3.2. Optimization of the Flavor Enhancement Technology

The flavor enhancement technique was used to create the enzyme hydrolysate (the Maillard process). Glucose was selected for reducing sugar by the Maillard process. The temperature (60 °C, 70 °C, 80 °C, 90 °C, and 100 °C), sugar concentration (0.5%, 1%, 1.5%, 2%, and 2.5%), and heating duration (40 min, 50 min, 60 min, 70 min, and 80 min) were the optimization conditions for the single factor optimization. The indicators were the absorbance at 420 nm and the results of the artificial sensory evaluation. All the experiments were repeated three times.

### 2.4. Preparation of the Samples for Flavor Analysis

The flavor analysis consisted of two groups of samples, with three samples in each group: (1) a certain amount of raw materials (crab legs and bodies) were crushed and added with deionized water to obtain a homogenized liquid (HO); (2) an enzymatic hydrolysate solution (EHS) was obtained by enzymolysis homogenate; (3) an Maillard reaction solution (MRS) was prepared by enzymolysis product of Maillard reaction.

### 2.5. E-Tongue

The taste analysis system (TS-5000Z, Insent Intelligent Sensor Technology Inc., Japan) is an analytical device that closely resembles human taste. It is equipped with a sensor probe for various tastes, including bitter flavors (SB2C00), and a reference probe. At 25 °C, the sensor measurement is automatically performed. The method is particularly sensitive to aftertaste and can objectively assess the sample’s salty, sour, bitter, astringent, and sweet flavors [11]. The HO, EHS, and MRS from the CLs and CBs were among the samples, and each sample was measured 5 times. SPSS 22.0 software was used to analyze the flavor principal component analysis (PCA), and Origin 8 software was used to draw flavor radar plots.

### 2.6. Free Amino Acid (FAA)

The hetero protein was eliminated using Trichloracetic acid (TCA) [12]. After the hetero protein was eliminated, the material was concentrated until it was dry. Hydrochloric acid was added to adjust the volume. Following filtration, an amino acid analyzer was used to determine which free amino acids were contained in the filtrate (Sykam S-433D, Sykam GmbH, Germany).

### 2.7. E-Nose

First, 3.5 mL of a sample solution was concentrated for 6 min in a 20 mL headspace bottle. According to an electronic nasal instrument (PEN3, Airsense, Germany), every sensor listed in the Appendix A, was utilized. The sample was measured when the response signal re-obtained its baseline value. The test parameters were 100 s for self-cleaning, 100 s for the sample collection, and 600 mL per min for the carrier gas flow rate. The ratio of conductance, G0/G or G/G0, was used to represent the gathered data. While G0 represented the conductivity before exposure to the gas sample, G0 represented the conductivity after.

The sample was measured when the response signal reached the baseline. The test’s settings were a carrier gas flow rate of 600 mL/min, a self-cleaning duration of 100 s, and a sample collecting period of 100 s. The gathered information was displayed as the conductance G0/G or G/G0 ratio. As opposed to G0, which reflected the conductivity before exposure to the gas sample, G was the conductivity of the sensor after exposure.

The PEN3 electronic nasal instrument’s Winmuster (version 1.6.2) software was used for the data analysis to implement the principal component analysis and linear discriminant analysis (LDA).

### 2.8. HS-SPME-GC/MS

HS-SPME-GC/MS analysis was carried out using an Agilent 7890-5975c gas chromatograph equipped with an HP-INNOWAX column (30 m × 250 µm × 0.25 µm) coupled to an automatic solid-phase microextraction system and mass spectrometer, as previously reported [13]. For the sample, 10 mL was placed in a headspace vial (20 mL), and 1 g of NaCl was gently mixed over the cap. After incubating at 60 °C for 40 min, the sample was extracted with a divinylbenzene/carboxy/polydimethylsiloxane (DVB/CAR/PDMS) SPME fiber (Supelco, Bellefonte, PA, USA) and desorption at 250 °C for 5 min.

The oven’s temperature was maintained at 40 °C for 5 min before being raised by 8 °C/min to 250 °C. The carrier gas, helium, was introduced at a flow rate of 5 mL/min, and 250 °C was maintained on the transfer line.

The mass spectrometer was run at 250 °C for the ion source temperature and 70 eV for the ionization voltage. To identify the volatile compounds, their mass spectra were compared to the NIST08.L database, and they were not adopted until the matching degree was above 80%. Utilizing the peak area normalization, the chosen chemicals were quantified.

### 2.9. Statistical Analysis

Statistical analyses of all experiments were performed in triplicates, and the data are presented as the mean ± standard deviation (SD). The statistical analysis was conducted with SPSS Statistics V22.0 (IBM Corporation, Armonk, NY, USA), and a *p* < 0.05 was significant. Graphs were drawn using Origin 8 software (OriginLab, Northampton, MA, USA).

## 3. Results and Discussion

### 3.1. Basic Components

The analysis of the fundamental properties of raw materials is based on the basic components of those materials. Table 1 illustrates the essential components of the legs and body of the Chinese mitten crab. The CLs and CBs exhibited moisture contents of 50.60% and 58.28%, respectively. On a dry weight basis, the ash content of the CLs and CBs was 46.52% and 41.30%, which may be due to the rich contents of chitin, calcium, and other minerals. Based on the dry weight, the crude protein content of the CLs (42.37%) was higher than that of the CBs (22.43%), while the chitin content of the CLs (8.01%) was significantly lower than that of the CBs (19.91%). The crude fat content of CLs and CBs was 1.26% and 5.30%. So, to sum up, the CLs and CBs typically had higher ash contents and lower moisture contents and were rich in mineral elements, proteins, chitin, and lipids when compared to shrimp by-products, providing a theoretical foundation for the production and incorporation of the high-value biotransformation of crab by-products [14]. Previous research has found that the processing of Chinese mitten crab contains abundant flavoring amino acids, which are excellent raw materials for the preparation of seasonings [15].

### 3.2. Optimization of Enzymatic and Flavor Enhancement Technology

CLs and CBs were used as raw materials to make condiments from *Eriocheir Sinensis* processing by-products. Figure 1 displays the Maillard reaction and enzymatic hydrolysis optimization results. Two enzymes, the flavor protease and papain, were selected for the enzyme species screening, taking into account the cost of industrialization, enzymatic effect, and taste quality. While the amino nitrogen content of the flavor protease progressively increased and was higher than that of the papain enzymatic digest, the amino nitrogen content obtained from the hydrolysis of the two crab by-products reached its peak at 15 min of hydrolysis time and remained pretty much unchanged after 30 min. Related research, however, suggested that the flavor protease enzymatic digest might offer a better flavor [16]. Therefore, the flavored protease was adapt to the enzymatic hydrolysis of crab by-products. According to the content of amino peptide nitrogen in the sample, the pretreatment time of the crab legs was determined to be 40 min, and that of the crab bodies was 0 min. The optimum solid–liquid ratio was 1:3 g/mL for the CLs and 1:4 g/mL for the CBs, respectively.

To further determine the enzyme addition amount and enzymatic digestion time of the flavor protein hydrolase, the amino nitrogen content decreased sharply when the enzyme’s additional amount increased to 5%, which might be due to the poor enzyme hydrolysis effect caused by the high enzyme concentration, and, thus, the enzyme addition amount was determined to be 4% for both. With longer enzymatic digestion times, the crab by-products’ amino nitrogen content gradually increased, along with their TVB-N level. After 9 h of enzymatic digestion, both the TVB-N concentration in the crab legs and body increased sharply, and the fishy odor also raised noticeably. The difference in the enzymatic digestion time between 6 h and 9 h was not significant, but the TVB-N quantity and fishy odor dramatically decreased. Consequently, the 6 h enzymatic hydrolysis time was applied to both the CLs and CBs.

The technical conditions of the flavor enhancement were optimized. Based on the experimental conditions and the energy cost of industrialization, the maximum temperature of the Maillard reaction was established at 100 °C. Below 100 °C, the sauces made from crab legs and bodies had either a heavy or mild fishy odor; however, at 100 °C, the fishy odor was nearly unnoticeable. The sauces made from crab legs and bodies had reaction times of 70 min and 60 min, and, at a 2.5% addition, they tasted strongly caramelized.

### 3.3. E-Tongue Analysis

The E-tongue is a qualitative analysis technique that classifies or identifies samples based on the composition of the taste sensor array [17]. A large rise in savory, freshness, and flavor richness, just a little increase in astringency, astringent aftertaste, and bitter aftertaste, and a decrease in bitterness and sourness, were all noticeable alterations in the sauce of the CLs compared to the homogenate, as shown in Figure 2. In comparison to the homogenate, the sauce made from the CBs underwent significant changes, which included a significant increase in the saltiness, freshness, and flavor richness, a slight increase in astringency and its aftertaste, as well as an increase in the bitterness and bitterness itself, and a decrease in acidity. This demonstrates that the Maillard reaction reduced the negative odors released by enzymatic digestion while imparting additional distinctive aromas. Meanwhile, the enzymatic digestion technique released more flavor-presenting chemicals. In addition, the saltiness, richness, and umami of the CL-EHS, CL-MRS, CB-EHS, and CB-MRS were significantly improved at the same time, compared with those indicators of the CL-HS and CB-HS. This phenomenon may indicate that these three taste patterns have mutually reinforcing effects.

Here, the sample flavor attributes were categorized using a principal component analysis (PCA) [18]. The total ratio of both principal component analyses for the crab legs and body was 98.83%, with PC1 (93.00%) and PC2 (5.83%). The CB-HS and CL-HS are separated in the first and fourth quadrants, showing that these two raw materials, without processing, have different flavor patterns. After processing, the CB-EHS and CB-MRS were in the second quadrant, while the CL-EHS and CL-MRS were in the third quadrant. These results indicated that enzymatic hydrolysis or enzymatic hydrolysis decoupling with the Maillard reaction could significantly change the flavor pattern of the raw materials. However, the characteristic flavor of the raw materials will still be retained after processing so as to obtain sauces with different flavor characteristics.

Because the PCA of both the crab legs and crab bodies was greater than 90%, the two sauces created from crab legs and bodies were excellent and flavorful, and the Maillard reaction and enzymatic aromatization processes that were used in this process were both beneficial.

### 3.4. FAA Evaluation

Free amino acids are a critical component of a substance’s flavor. Je et al. (2005) investigated the free amino acid content of fermented oyster sauce at various times and discovered that the content might enhance the fermentation of the oyster sauce’s flavor [19]. Table 2 presents the FAA values from the samples. There were four, six, and seventeen FAA types of HO, EHS, and MRS obtained from CLs; there were seven, seventeen, and seventeen FAA types of HO, EHS, and MRS obtained from CBs. In the Maillard reaction, some of the amino acids participated in its reaction as precursors of the reaction, resulting in a reduction in its content. However, this part of the amino acids will be converted into other amino acids to continue to affect the flavor of the enzymatic solution [20]. The total free amino acid content in the CLs’ HO was 0.16 mg/mL, while it rose to 0.89 mg/mL in their MRS. The total free amino acid content in the HO of the CLs was 0.19 mg/mL, but through enzymatic hydrolysis and the Maillard reaction, it rose to 0.48 mg/mL in the MRS of the CLs. The total amounts of umami, sweet, and bitter amino acids were 0.06 mg/mL, 0.51 mg/mL, and 0.55 mg/mL in the sauces from the CLs, respectively, and 0.06 mg/mL, 0.25 mg/mL, and 0.32 mg/mL in the sauces produced by the CBs. After the Maillard reaction, the bitterness of the enzymolysis solution could be significantly improved, and the umami taste was improved, which was consistent with Yan et al.’s (2021) study [21].

Additionally, the CL and CB sauces had high concentrations of alanine and histidine, which are thought to be important elements in the flavor and taste of CLs and CBs, respectively.

### 3.5. E-Nose Analysis

The conductivity graphs of the six samples used in this experiment are displayed in Figure 3. Due to the various kinetic properties of the sensor, the data resulting from the interaction of the volatiles on the detecting element are transformed into an odor fingerprint.

Principal component analysis (PCA) and linear discriminant analysis (LDA) are widely used in E-nose data analysis. Figure 4 displays the E-nose analysis’s findings. During the use of reduced dimensionality, the PCA analysis first evaluated the similarity between the clusters before visualizing all the data in the dataset [22]. The PCA value of the CBs was 96.07%, of which PC1 and PC2 accounted for 70.84% and 25.23%, respectively. The PCA value of the CLs was 98.83%, with PC1 and PC2 accounting for 97.71% and 1.12%. Indicating that both types of volatile chemicals accounted for the majority of the samples, the PCA of the crab legs and bodies was higher than 85%. The distance between the sample points demonstrated that they were far apart and highly different, and the primary two scent component types and quantities varied noticeably between the samples.

LDA is a technique for determining the projection direction, d, between various sample types [23]. The LDA value of the CLs was 98.78%, with LDA1 and LDA2 accounting for 80.23% and 18.55% of that value, respectively. The LDA value of the CBs was 99.93%, with LDA1 and LDA2 standing for 93.30% and 6.63%, respectively. The samples were well-clustered, the LDA values of the crab legs and bodies were greater than 85%, and the enzymatic digestion method and the Maillard reaction enhancement process, which were employed in the creation of the sauces from the crab legs and bodies, were both practically effective.

### 3.6. HS-SPME-GC/MS Analysis

The carotenoid-as-precursor reaction pathway, the lipid-as-precursor reaction pathway, the glycoside-as-precursor reaction pathway, and the Maillard reaction pathway are the four main pathways for the power generation of flavoring substances [24]. Figure 5 depicts the total ion flow chromatograms (TICs) of the volatile components. The identification of the peaks in the TIC of the tested samples via a NIST search enabled the identification of specific volatile compounds in the samples, as shown in Table 3. The number of MRS species from the crab body and leg sources changed significantly when compared to the number of species from the HO and EHS, and a huge percentage of the effective volatile components belonged to aldehydes, ketones, alcohols, and pyrazines. This demonstrates that, following treatment with enzymatic digestion technology and Maillard aroma enhancement technology, the species and quantity of volatile chemicals changed. During the enzymatic and flavor-enhancing treatments, the types and concentrations of volatile chemicals in the seasonings of the crab legs and bodies were changed dramatically. The types of volatile compounds detected were 11, 17, and 22 in the HO, EHS, and MRS obtained from the CLs, and 15, 21, and 13 in the HO, EHS, and MRS obtained from the CBs, respectively. Aldehydes, ketones, alcohols, or pyrazines accounted for the majority of the volatile substances found in the samples.

Trimethylamine is an essential sign for determining if aquatic items are still fresh. Trimethylamine tends to emit a strong fishy stench and has a low odor threshold. Trimethylamine substances were discovered in the HO homogenate of the crab legs and bodies, though, after the treatment with enzymatic technology and Maillard aroma enhancement technology, no additional trimethylamine substances were found. This may be related to the antioxidant capacity of the Maillard reaction, which effectively reduces the fishy taste [25].

The total proportion of aldehydes greatly rose in the CL and CB enzymatic hydrolysates compared to the homogenate, whereas the number of ketones and alcohols were massively reduced. The primary fishy components in fresh whitefish are C8- and C9-alcoholic compounds, such as 1-octen-3-ol [26]. The EHS of the CLs and CBs contained 2.94% and 3.52%, respectively, of C8- and C9-alcoholic chemicals. As a result, the CL and CB EHSs require extensive consideration.

In comparison to the homogenate, both the crab leg and body sauces contained a higher proportion of aldehydes. Generally speaking, most aldehydes have aromatic, fruity, or other pleasing scents [27]. Benzaldehyde is one of the chemicals that gives CL and CB sauces their distinct flavors, and it contributes more than half of the entire odor of these sauces. A common food-grade flavor, benzaldehyde has an almond flavor. Cai et al. (2016) also found benzaldehyde to be a contributor to the specific aroma of shrimp hydrolysis products via the Maillard reaction [28]. The sauces for the crab legs and bodies contained a total of seven pyrazines, six of which were new compounds. Pyrazines were the major contributors to the nutty/roast and basic meaty aroma in the Maillard reaction products [29]. Pyrazines are frequently utilized as natural food-grade antioxidants and anti-browning agents. They also have a powerful roasted and nutty flavor [30]. They have a good antioxidant capacity, which is why they are utilized as antioxidants and anti-browning agents. The dissolution of free radical chains, the breakdown of hydrogen peroxide, the elimination of reactive oxygen species, and the chelation of metal ions are several instances of antioxidant mechanisms [31]. Liu et al. (2015) adopted the two-step process of enzymolysis and the Maillard reaction to prepare the Maillard product of an oyster meat hydrolysate with an antioxidant capacity [32]. According to Negroni et al. and Zamora and Hidalgo et al.’s research, the different fatty acid contents of these two crab processing by-products are associated with the varying quantities of pyrazine in CL and CB sauces [33,34]. Due to low sensory thresholds, sulfur- and nitrogen-containing compounds accounted for a small percentage of the volatile compounds in the CL and CB sauces, at 2.43% and 8.58%, respectively, and both were significant flavoring agents for supplying a meat flavor, barbecue flavor, and beef flavor [35]. Alkanes and alcohols were present as well, but due to their low relative quantity or high odor thresholds, their contribution to the overall odor was insignificant.

## 4. Conclusions

A high-value technology for utilizing Chinese mitten crab processing by-products (CLs and CBs) was established in this study. Flavorful sauces could be obtained after enzymatic hydrolysis decoupling with the Maillard reaction-processing of CLs and CBs. The flavor protease was suitable for CL hydrolysis, while flavorase was chosen for the CBs. A glucose concentration of 2.5% is favorable for the following Maillard reaction. Compared with the OH and EHS, the amount and content of flavorful amino acids were significantly increased after the Maillard treatment, and the substances with bad flavors, such as amines and alcohols, were significantly decreased or even disappeared. Meanwhile, the characteristic flavor substances, such as aldehydes, pyrazines, and sulfur-containing compounds, were generated, which brought sweet and roast flavors to the MRS. The composite processing mode could obviously improve the flavor, remove the fishy smell, and also bring antioxidant activities, which provides a theoretical basis for the comprehensive utilization of crab resources.

## Figures and Tables

**Figure 1 foods-12-00051-f001:**
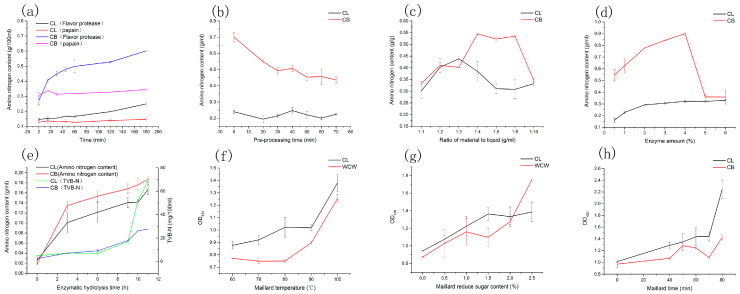
Ptimiziation of the enzymatic and Maillard process conditions: (**a**–**e**) Optimization of the enzymatic process conditions: (**a**) the choice of enzyme species, (**b**) pre-processing time, (**c**) ratio of material to liquid, (**d**) enzyme amount, and (**e**) time of enzymatic hydrolysis. (**f**–**h**) Optimization of the Maillard process conditions: (**f**) the Maillard temperature, (**g**) Maillard-reduced sugar content, and (**h**) Maillard time.

**Figure 2 foods-12-00051-f002:**
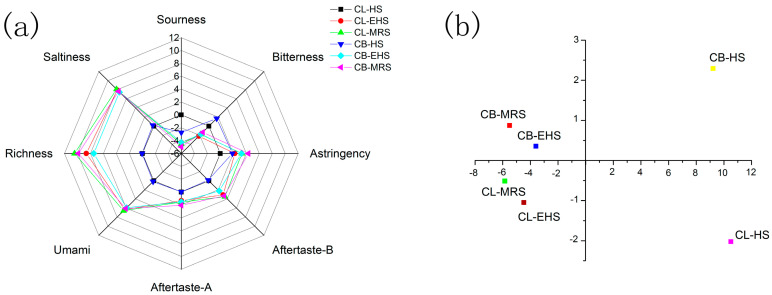
E-tongue patterns: (**a**) the taste polar pattern in the samples and (**b**) PCA analysis.

**Figure 3 foods-12-00051-f003:**
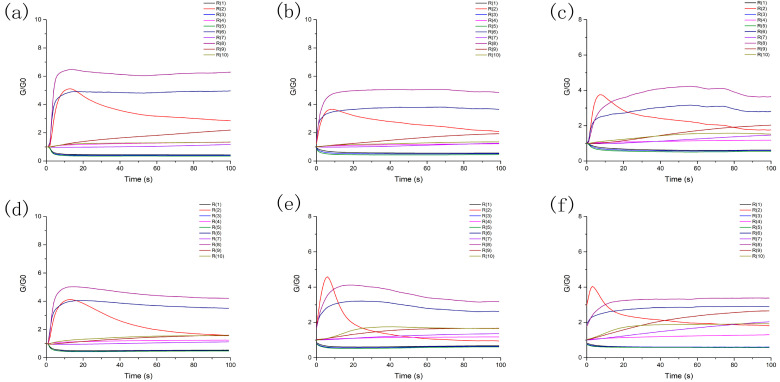
E-nose patterns: (**a**) ten sensors’ response to volatile components from HO of the CLs, (**b**) ten sensors’ response to volatile components from EHS of the CLs, (**c**) ten sensors’ response to volatile components from MRS of the CLs. (**d**) ten sensors’ response to volatile components from HO of the CBs, (**e**) ten sensors’ response to volatile components from EHS of the CBs, and (**f**) ten sensors’ response to volatile components from MRS of the CBs.

**Figure 4 foods-12-00051-f004:**
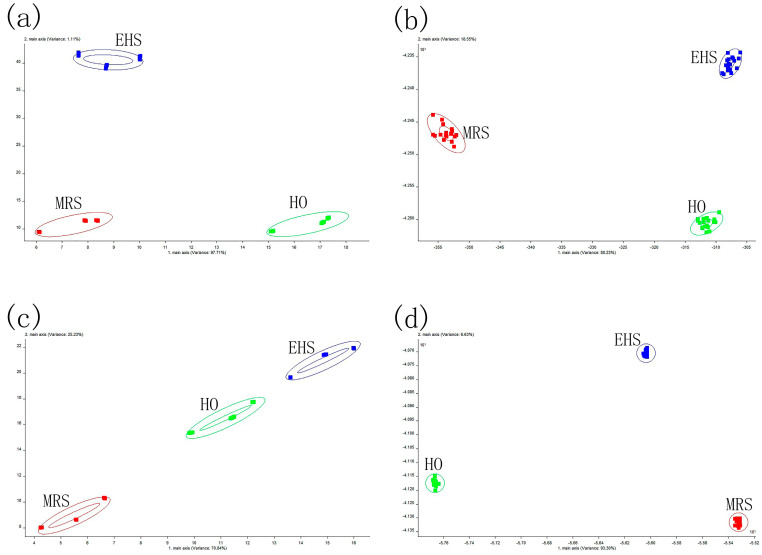
E-nose patterns: (**a**) PCA of crab legs, (**b**) LDA of crab legs, (**c**) PCA of crab bodies, and (**d**) LDA of the crab bodies.

**Figure 5 foods-12-00051-f005:**
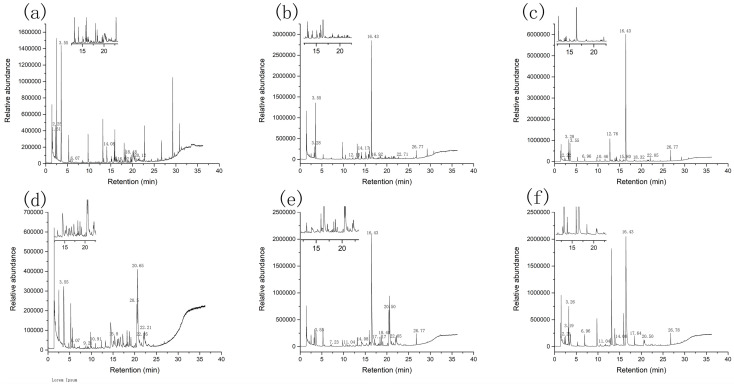
Total ion chromatogram of volatile components: (**a**) HO of crab legs, (**b**) EHS of crab legs, (**c**) MRS of crab legs, (**d**) HO of crab bodies, (**e**) EHS of crab bodies, and (**f**) MRS of crab bodies.

**Table 1 foods-12-00051-t001:** Basic components of CLs and CBs from the Chinese mitten crab.

	Moisture (%)	Ash * (%)	Crude Protein * (%)	Crude Lipid * (%)	Crude Chitin * (%)
CL	50.60 ± 1.16	46.52 ± 0.55	42.37 ± 0.18	1.26 ± 0.34	8.01 ± 0.75
CB	58.28 ± 0.27	41.30 ± 0.21	22.43 ± 2.09	5.30 ± 0.18	19.91 ± 0.92

Data are means ± SD; * Dry matter; the crab legs (CLs) and crab bodies (CBs).

**Table 2 foods-12-00051-t002:** Proportion of hydrolyzed amino acid composition, the concentration, and the increasing ratio of free amino acids.

Taste Attribution				Free Amino Acid					
		CL				CB			
		HO (mg/mL)	EHS (mg/mL)	MRS (mg/mL)	The Increasing Ratio (%)	HO (mg/mL)	EHS (mg/mL)	MRS (mg/mL)	The Increasing Ratio (%)
Proline	Sweet/bitter (+)	0.03	0.05	0.04	25.00	0.01	0.03	0.03	66.67
Aspartic acid	Umami (+)	0.00	0.04	0.02	100.00	0.00	0.04	0.02	100.00
Threonine *	Sweet (+)	0.00	0.05	0.02	100.00	0.00	0.03	0.01	100.00
Serine	Sweet (+)	0.00	0.07	0.03	100.00	0.00	0.04	0.01	100.00
Glutamic acid	Umami (+)	0.00	0.09	0.04	100.00	0.01	0.09	0.04	75.00
Glycine	Sweet (+)	0.04	0.10	0.05	20.00	0.05	0.08	0.03	−66.67
Alanine	Sweet (+)	0.05	0.18	0.11	54.55	0.05	0.09	0.05	0.00
Cysteine	Bitter/sweet (−)	0.00	0.00	0.02	100.00	0.00	0.02	0.01	100.00
Valine *	Sweet/bitter (−)	0.00	0.09	0.07	100.00	0.00	0.05	0.03	100.00
Methionine *	Bitter/sweet (−)	0.00	0.04	0.02	100.00	0.00	0.02	0.01	100.00
Isoleucine *	Bitter (−)	0.00	0.06	0.05	100.00	0.00	0.04	0.02	100.00
Leucine *	Bitter (−)	0.00	0.13	0.09	100.00	0.00	0.07	0.04	100.00
Tyrosine	Bitter (−)	0.00	0.07	0.04	100.00	0.00	0.03	0.02	100.00
Phenylalanine *	Bitter (−)	0.00	0.09	0.05	100.00	0.00	0.04	0.02	100.00
Histidine *	Bitter (−)	0.00	0.05	0.09	100.00	0.01	0.03	0.07	85.71
Lysine *	Sweet/bitter (−)	0.00	0.12	0.06	100.00	0.01	0.08	0.04	75.00
Arginine	Sweet/bitter (+)	0.04	0.17	0.09	55.56	0.05	0.08	0.03	−66.67
EAA/NEAA	--	0.00	53.85	53.45	--	5.56	33.72	17.44	--
EAA/TAA	--	0.00	35.00	34.83	--	5.26	66.28	38.37	--
Total	--	0.16	1.40	0.89	--	0.19	0.86	0.48	--
Umami	--	0.00	0.13	0.06	--	0.01	0.13	0.06	--
Sweet	--	0.16	0.87	0.51	--	0.17	0.52	0.25	--
Bitter	--	0.07	0.87	0.55	--	0.08	0.49	0.32	--

* Essential amino acids of crab waste: threonine, methionine, valine, histidine, Isoleucine, leucine, lysine, and phenylalanine. EAA means essential amino acids. NEAA means nonessential amino acids. TAA means total amino acids. Taste attributes (+ = pleasant, − = unpleasant).

**Table 3 foods-12-00051-t003:** Composition of volatile compounds detected in the HO, EHS, and MRS of CLs and CBs from *Eriocheir Sinensis*.

			CL			CB		
			HO (%)	EHS (%)	MRS (%)	HO (%)	EHS (%)	MRS (%)
Aldehydes	Retention Time (min)	Description of Odor	ND	63.55	67.55	ND	55.07	74.68
2-methyl butyraldehyde	3.21	Chocolate		1.51	2.42		1.12	2.73
3-methyl butyraldehyde	3.28	Chocolate		4.00	7.14		3.15	7.14
Isobutyraldehyde	2.21	a pungent odor						1.31
Decanal	15.99	Buckwheat flavor		3.70	0.34			
Benzaldehyde	16.43	Almond fragrance		51.31	55.15		50.41	63.50
2-methyl-propanal	2.22	Wet grain flavor			0.68			
Benzeneacetaldehyde	18.32	Sweet			0.43			
Nonanal	14.17	Strong oily smell and sweet orange		2.31	1.39			
Hexanal	7.23	a pungent odor					0.39	
Octanal	12.16	Fruity aroma		0.72				
**Ketones**			**18.45**	**1.07**	**ND**	**13.80**	**3.25**	**ND**
Acetone	2.25	Aromatic smell	10.32					
Methyl hepten	13.11	Fresh green, citrus-like	0.60					
2-Nonanone	14.08	Fragrant	6.66	1.07		13.80	2.69	
2-Heptanone, 6-methyl-	11.04	Tobacco leaf					0.16	
3-Octanone	11.40	a pungent odor					0.23	
2-Octanone	12.17	Cooked shrimp					0.17	
2-Undecanone	17.64	Cream, cheese favor	0.87					
**Alcohols**			**56.33**	**30.92**	**9.08**	**68.88**	**28.68**	**7.69**
Ethanol	3.55	Special fragrance	51.87	23.03	8.34	21.58	8.36	7.22
1-Octanol	16.92	Metal, medicinal, Mushroom	0.99	0.69		3.88		
1-Nonanol	18.48	Citrus smell	3.47			3.47	3.52	
1-Hexanol	13.38	Green grass		2.26			0.48	
3-Methyl-1-butanol	10.57	a mild, choking alcohol odor					1.10	
1-Octen-3-ol	15.12	Mushroom flavor, Earthy taste		2.25				
2-Ethylhexanol	15.80	Sweet and light floral		2.21	0.26	3.32		
(Z)-dodec-4-en-1-ol	20.50	nd				9.37	12.60	0.47
(Z)-4-Decen-1-ol	20.65	a food additive				24.40		
9-Decen-1-ol	22.05	Fatty alcohols				2.86	2.62	
1-Undecanol	22.71	Citrus scent		0.48				
n-Tridecan-1-ol	22.72	Pleasant smell			0.48			
**Alkanes**			**ND**	**ND**	**ND**	**7.80**	**6.36**	**2.33**
Bromodicloromethan	9.21	By-product of drinking water				0.35		
Cyclohexane, 1,2,3-trimethyl-	20.24	nd				1.12		
Cyclooctene, 3-methyl-	22.21	nd				6.34	6.36	
Bicyclo [4.1.0]heptane, 3-methyl-	20.71	nd						2.33
**Aromatics**			**2.77**	**ND**	**1.60**	**0.57**	**0.83**	**2.24**
Toluene	6.07	Aromatic	1.32			0.57		
2,4-Di-tert-butylphenol	26.78	Special smell						2.24
Naphthalene	19.79	Mothball smell	1.45		0.26			
1,2-Epoxyoctahydropentalene	17.33	Mothball smell					0.83	
Butylated Hydroxytoluene	22.05	Odorless and tasteless			1.35			
**Pyrazines**			**ND**	**0.78**	**16.23**	**ND**	**0.45**	**4.33**
2,5-Dimethyl pyrazine	12.76	A pungent, fry aroma and chocolate, creamy		0.78	9.98		0.45	3.45
Pyrazine	10.46	Aromatic			1.18			
2-Methylpyrazine	11.62	Nutty, roasted			1.19			0.88
Ethylpyrazine	13.00	Nutty, roasted, meaty, Fishy			0.36			
2-ethyl-5-methyl-pyrazine	14.08	Roast			0.49			
2,3,5-Trimethylpyrazine	14.31	Roast			2.12			
3-ethyl-2,5-dimethyl-pyrazine	15.04	Nuts, roasted scent			0.92			
**Phenols**			**ND**	**2.16**	**3.10**	**ND**	**2.27**	**ND**
2,5-Bis(1,1-dimethylethyl) phenol	26.77	Special smell		2.16	3.10		2.27	
**Esters**			**ND**	**0.82**	**ND**	**ND**	**2.57**	**ND**
Formic acid, octyl ester	17.17	Barbecue					2.57	
Pentanoic acid, 2,2,4-trimethyl-3-carboxyisopropyl, isobutyl ester	21.60	nd		0.82				
**Amines**			**6.53**	**ND**	**ND**	**7.88**	**ND**	**ND**
Trimethylamine	1.51	Smelly smell, amine smell	6.53			7.88		
**Terpenes**			**ND**	**0.69**	**ND**	**ND**	**ND**	**ND**
Azulene	19.81	Mushroom incense		0.69				
**Organosulfur compounds**			**ND**	**ND**	**2.43**	**ND**	**0.25**	**8.58**
Dimethyl disulfide	6.96	a cabbage-like odor			1.20		0.25	2.54
Dimethyl trisulfide	13.91	Sulfur, leek, cooked onion			1.23			6.04
**Oxime s**			**15.91**	**ND**	**ND**	**ND**	**ND**	**ND**
Oxime-, methoxy-phenyl-	20.12	nd	15.91					
**Pyrroles**			**ND**	**ND**	**ND**	**0.45**	**ND**	**ND**
N-Methyl pyrrole	8.75	Roast				0.45		
**Furans**			**ND**	**ND**	**ND**	**0.61**	**0.26**	**ND**
2-Pentylfuran	10.91	Bean, fruit, earth, green				0.61	0.26	
**Pyrimidines**			**ND**	**ND**	**ND**	**ND**	**ND**	**0.15**
Pyrimidine	10.46	Aroma						0.15

ND = no detection.

## Data Availability

Data is contained within the article.

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
