# Peer review of "Preparation and Characterization of Flavored Sauces from Chinese Mitten Crab Processing By-Products"

_foods, 2022, doi:10.3390/foods12010051_

Round 1

Reviewer 1 Report

The manuscript is written with clear understanding of the project addressed. However, there are major concerns that need to be addressed to enhance the quality of the manuscript. My specific comments are as follows:

Abstract:

Add main finding of the study

Introduction:

“The Chinese mitten crab (Eriocheir Sinensis), which is primarily found in North Asia and the Thames Valley region, is a significant freshwater economic crab in China.” Add citation

Discuss literatures on mailard reaction on aquatic byproduct

Based on your objectives, please compare how your study is different from those that have already been published

Methods:

CL and CB? Spell out the acronym

How many samples are used? Explain briefly

Justify how the samples were selected/obtained

Add data analysis part, eg. Statistical analysis involved

Results and Discussion:

Discuss whether there are significant difference between those two samples in terms of basic components

What are the correlation of taste pattern based on e-tongue result

Based on classification performance of PCA, what can be deducted from the finding?

Instead of mentioning the results, the authors should justify/explain the findings

 Conclusion:

Add on main finding/results of the study. What are the main outcome based on the results. The authors should highlighted this matter

General comments:

Please check the reference styles and grammar of the manuscript.

Reviewer 2 Report

Comments are given in the attached file.

Reviewer 3 Report

The submitted article is an interesting investigation of a novel, effective approach to using crab processing by-products, increasing their overall worth, and offering theoretical backing for the high-value utilization of crab sauces. The work is well written, and clearly presented; however, it is my suggestion to include statistical analysis in the optimization of the enzymatic process condition and compare results with the obtained TVB-N results.  Same minor specific issues also need the author's attention:

Page 2, highlight how many repetitions analyzes were performed.

Page 6, please, state the program used for PCA analysis and provide variable contribution values.

Page 8, Figure 3 caption is misplaced; please check.

Pages 15-16, some references are not prepared according to the journal requirements; please check.

Round 2

Reviewer 1 Report

The authors have addressed all the comments. Hence, the paper can be accepted.